# Measuring the Degree of Balance between Urban and Tourism Development: An Analytical Approach Using Cellular Data

**Cheng Shi [1,2], Mengyang Liu [3] and Yu Ye [1,2,\***

1   College of Architecture and Urban Planning, Tongji University, Shanghai 200092, China;
    chengshi@tongji.edu.cn
2   Key Laboratory of Ecology and Energy-Saving Study of Dense Habitat, Tongji University,
    Shanghai 200092, China
3   School of Architecture and Urban Planning, Huazhong University of Science and Technology, Wuhan 430073,
    China; mengyang_liu11@126.com
*   Correspondence: yye@tongji.edu.cn; Tel.: +86-182-1726-8257

**Abstract:** This study presents an analytical approach for measuring the degree of balance between urban and tourism development, which has been previously analyzed qualitatively and was difficult to measure. With the help of 1012 million cellular data records generated by 20 million users in two weeks, we tracked the behavior of residents, commuters, and tourists at a set of historical conservation areas in central Shanghai. We calculated the degree of balance and visualized it via ternary graphs. Moreover, the relationships between key urban features derived from multi-sourced urban data and balanced degrees of tourism development were analyzed via multinomial logistic analysis. Insights gained from this analysis will help to achieve a more scientific decision-making process toward balanced urban development for historical conservation area. Achievements in this study contribute to the development of human-centered planning through providing continuous measurements of an "unmeasurable" quality.

**Keywords:** balanced tourism; cellular data; built environment; multi-sourced urban data; Shanghai





## 1. Introduction

Tourism has been proven to be a key contributor of economic growth in many countries [1,2], especially in developing or newly developed countries with a rising population and governmental economic goals. The development of tourism can motivate the optimization of the city image including the re-habitation of cultural heritage and natural environment. In addition, socio-economic issues like employment opportunities [3], recreation, and cultural activities [4] are promoted as well. However, tourism can bring adverse effects through congestion and cultural distance problems between tourists and locals [5,6]. Inefficient land use or urban planning can also a result of over-proliferated tourism [7].

Therefore, balanced urban tourism is considered the ideal vision for future urban tourism development, mitigating those conflicts and enhancing the benefits to the locality, which is also an enduring topic in urban and tourism development research. The common goals of economic, sociocultural, and resource sustainability and cultural heritage protection should be shared by different stakeholders including the community, the tourism industry, and local residents [8]. Considering that the development of tourism is a dynamic process between supply and demand, the exploitation of tourism facilities may directly activate tourism demand and trigger potential influences on related aspects to the local area [9]. Therefore, to guarantee balanced development, especially for areas with abundant tourism resources, urban and tourism management experts need to evaluate the dispersal of tourists and local residents appropriately.

As a response to this demand, Global Positioning System (GPS) tracking devices has been applied to acquire the positioning records of tourists and local residents [10,11]. This

tracking technique helps to reveal many new insights. Nevertheless, the distribution of tracking devices is a time-costly and low efficiency process. It is also difficult to collect large samples within city scale. Therefore, cellular data with the capacity of recording spatiotemporal distribution of human behavior might bring new research potentials for measuring the degree of balance between urban and tourism development.

In this context, this paper is attempting to develop an analytical approach with the help of multi-sourced urban data for enabling urban planners and tourism managers to evaluate the degree of "balance", especially for areas with tourism resources, which could provide a more evidence-based decision-making process towards balanced urban development. Hence, to test the effectiveness of introducing new urban data, the data will be tested in research sites following the six sections. After the introduction, Section 2 introduces background research, and Section 3 explains the methodology of cellular and other urban data process. Section 4 describes the results on balancing tourism and different categories of urban issues. Then, in Section 5, the results will be discussed in order to answer the research questions. Suggestions for balanced urban development for historical conservation areas, limitations, and future research are also mentioned.

## 2. Literature Review

### 2.1. Measuring the Degree of Balance between Urban Development and Tourism Planning: Previous Attempts and Difficulties

To monitor the impact of tourism on local communities, Faulkner and Tideswell [12] propose a conceptual framework, including the ratio of tourists to local residents as an important indicator. Higher value of this ratio means a high likelihood of disturbing local residents regardless of the actions of tourists. This concept mainly focuses on tourists in relation to local residents and has also been adopted in many other studies [13].

Nevertheless, it is hard to get a big picture on distribution and behavioral characteristics of tourists to get an indicator of balanced development, although direct and indirect explorations have been made. Cros [14] inspects tourism congestion through interviews. Kearsley and Coughlan [15] attempt to clarify the tourism behavior mechanisms through questionnaire surveys. These two approaches obtain samples which are too limited to draw a convincing conclusion. The majority of researchers use the conventional official statistic of EU tourism provided by Eurostat [16] and the World Tourism Organization on a yearly basis, which has much more sample-coverage but limited spatiotemporal details.

Moreover, the tourist–resident ratio has the limitation of excluding other stakeholders. As Waligo, Clarke, and Hawkins [17] revealed, sustainable tourism needs to consider more stakeholders from industries, the government, etc. Moyle et al. [18] also argued that the concept of tourism impact could be discussed in various scales, referring to the long-term or short-term, and individual or accumulative influences of continuous interactions between tourists and local communities of tourist destinations, or local enterprises. Hence, there are other groups which could be involved in this research on balanced tourism, such as commuters who play an important role in supporting both tourism activities and local residents. Further explorations in this direction are still needed.

### 2.2. New Research Potentials Generated from Emerging Multi-Sourced Urban Data

Batty [19] emphasizes the transition of understanding cities from a state to a multi-dimensional system. It is essential to capture data on tourists with all the geographic and temporal variations in a more efficient way [20]. Emerging data accompanying the improvement of information and communication technologies have opened up new research potential that could not be achieved with conventional approaches [21,22]. Specifically, social media data can be used to generate international destination patterns [23]; web search engines can forecast travel demand [24]; and GPS data can provide high resolution spatial and temporal data [25]. Edwards and Griffin [26] tracked 154 participants to see how tourists moved by using GPS devices within Melbourne and Sydney. Li et al. [11]

applied GPS data to detect the spatial and temporal distribution of tourists as an indicator of balanced tourism development.

Among these new urban data, cellular data may play an important role as these enable a graphic representation of the intensity of urban activities and their evolution through space and time [27]. The advantage of cellular data in identifying tourists' pattern has been noticed as it could provide broad geographic and temporal coverage of datasets with their behavior pattern at fined-grained resolution [10,28,29]. Benefiting from high cell-phone-penetration rate, cellular data have a large sampling rate which is close to a full representation of the whole population. Its unique random phone ID enables researchers to recognize different group interactions through their peculiar pattern of trip frequency and origin–destination [30], which enables a capability of identifying different profiles of local residents from temporary population such as commuters and visitors [31]. In this context, cellular data has been applied to optimize tourism development via identifying tourists' group movement patterns [32], measuring tourists' spatiotemporal preference on destination visiting [33]. It is also interesting to apply cellular data to identify the effect of tourists' party size on their tourism behavior [34]. In short, the advantages of cellular data have been well recognized, which also bring research potential for analyzing the behavior of different stakeholders besides tourists to figure out the mechanism of balanced tourism development.

## 3. Materials and Methods

### 3.1. Research Questions and Framework

To promote a more balanced urban development for areas with abundant tourism resources, this study attempts to develop a quantitative analytical approach with the help of cellular data to measure the degree of "balance" and inform better urban management strategies. Our two main research questions are:

(1) How can the degree of "balance" in urban areas based on the dispersal of human flow be evaluated?

(2) How do different categories of urban features affect balanced urban development?

The analytical framework illustrated in Figure 1 contains four phases. First, cellular data were collected and cleaned while the research sites were set. Meanwhile, the points-of-interests (PoIs) representing urban facilities and built environment data recording urban planning indices were crawled via Baidu Map API. After that, cellular data and PoIs and built environment data were processed separately. The positioning distribution of residents, commuters, and tourists within the selected research sites was computed via cellular data from a mobile phone network. Meanwhile, key urban features, including land use and urban morphological factors, were calculated based on the collected PoIs and built environment dataset. Based on this, we were able to visualize the degree of balance between tourism and urban development using ternary graphs. Then, we ran a multinomial logistic regression in order to explain the impact of various urban features on balanced development of specific research sites.

### 3.2. Research Cases

Shanghai was chosen as an example of a historical city which has undergone fast urbanization. As one of the largest and most developed cities in China, tourism development is one of the main factors to promote economic growth in Shanghai. The total income of tourism and related services in 2018 was 50.9 billion CNY (Chinese Yuan), occupying 6.4% of the GDP of Shanghai [35]. Fast tourism development has brought pressure on historical areas. In this context, Shanghai Resources Bureau [36] has published a conservation agenda for twenty historical conservation areas (with twelve of them located in the city center) to promote a more sustainable balance between tourism and urban development.

Twelve urban historical conservation areas in Shanghai city center were selected for this research, as shown in Table 1, and they are divided into 25 subsets according to current zoning settings and land-use occupancy. Another eight suburban historical

conservation areas were also included. Their neighborhood features and spatial distribution are illustrated in Figure 2. Specifically, Laochengxiang and Longhua Road are old Shanghai town center, which are famous for ancient temples. The Bund and the Hengshan-Fuxin are British and French concessions in the 20th century respectively, and both of them are successful cases of re-adaption for commercial use. Jiangwan is a cultural district where historical universities and institutions are located. Others belong to the category of public concession with many garden villas. Eight suburban areas are traditional Chinese Watertown. This versatile history and land use lead to different morphological features for each historical conservation area. However, they still have a common feature in that all of them are attractive to both foreign and domestic tourists.

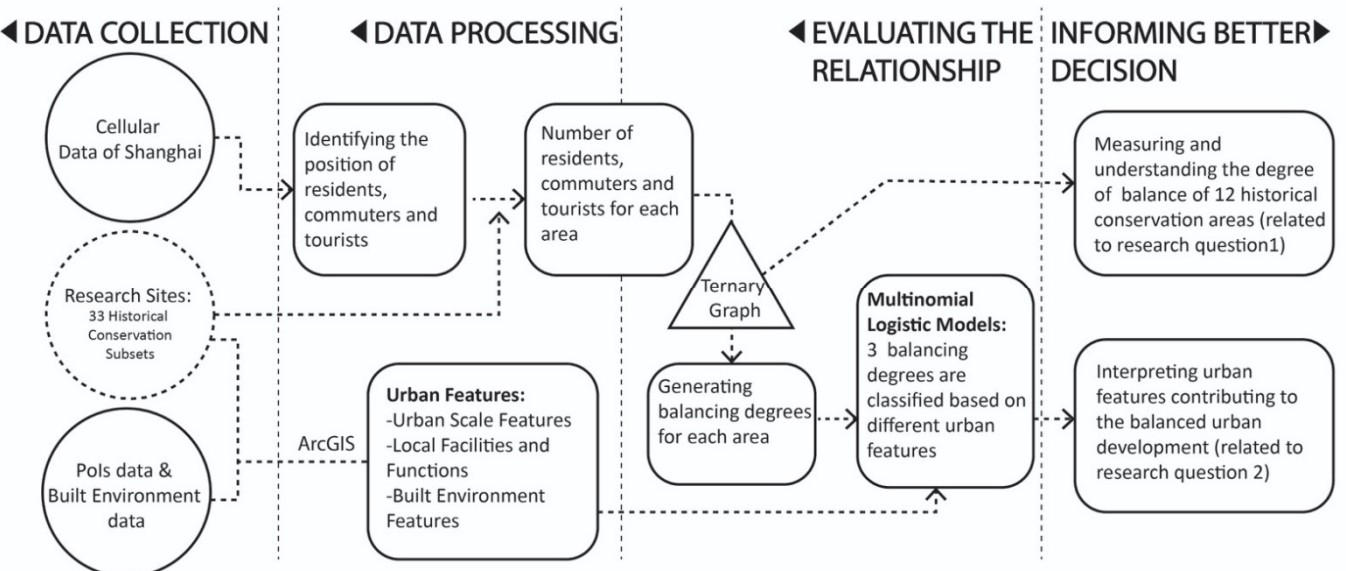

**Figure 1.** Analytical framework.

**Table 1.** List of 20 Shanghai historical conservation areas and 33 subsets.

| Name | Subsets | Construction Time |
| --- | --- | --- |
| Hengshan-Fuxin | HF-01/02/03/04 | 1919–1941 |
| Hongqiao Road | HQ-01/02/03 | 20th Century |
| Jiangwan | JW-01/02/03 | 1930–1940 |
| Laochengxiang | LC-01/02/03/04 | 16th Century |
| Longhua Road | LH-01 | 15th Century |
| Renmin Square | RM-01 | 1860–1941 |
| Nanjingxi Road | NJ-01/02 | 1899–1941 |
| Shanyin Road | SY-01/02 | 1900–1925 |
| Tilanqiao | TL-01 | 1939–1945 |
| The Bund | WT-01 | 1900–1941 |
| Xinhua Road | XH-01 | 1925–1941 |
| Yuyuan Road | YY-01/02 | 1919–1941 |
| Pudong Gaoqiao | PG | 10th Century |
| Pudong Xinchang | PX | 10th Century |
| Minhang Qibao | QB | 11th Century |
| Xujin Panlong | PL | 16th Century |
| Jiading Nanxiang | NX | 6th Century |
| Sijing Guzhen | SJ | 10th Century |
| Hangtou Xiasha | XS | 8th Century |
| Songjiang Fucheng | FC | 13th Century |

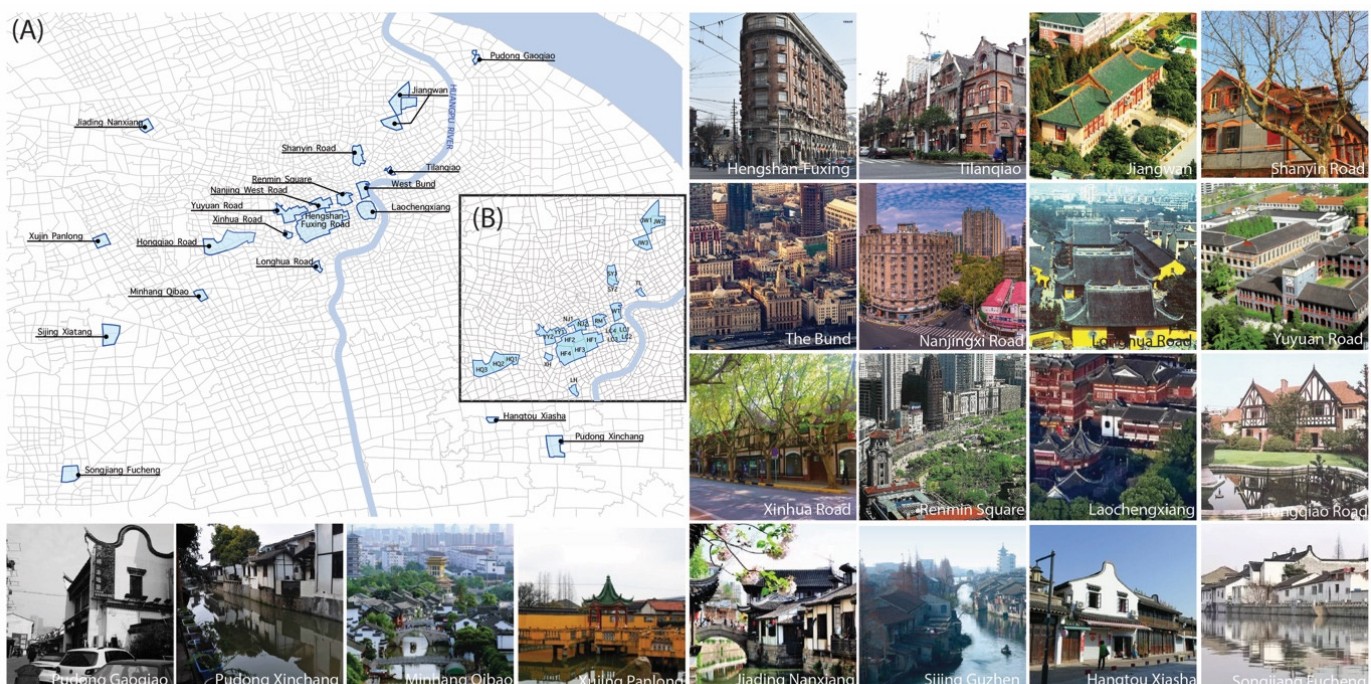

**Figure 2.** (**A**) Location map of the Shanghai historical conservation area. (**B**) Subsets from the historical conservation area in Shanghai city center.

### 3.3. Cellular Data Processing

The number of positioning points for tourists, commuters, and local residents were calculated using cellular data according to specific spatiotemporal patterns. The cellular database used herein was provided by China Mobile, an operator covering approximately 65% of mobile users in Shanghai [37]. This database consists of call detail records (CDRs) in Shanghai lasting 14 days in March 2018. Considering the balance between urban and tourism development in this study is a long-term pursuit which is not targeting peak days, we are mainly focusing on people's daily lives and how built environment features affect this issue. Therefore, we only involved the degree of balance in weekdays and weekends in the current study. In addition, positioning data recording the position of Internet of Things (IoT) has been moved out in the China Mobile's data preparation process. Considering these IoTs, e.g., shared bikes and electric metrics, usually obtain quite different movement routes, it is easy to identify the IoT records with the public's daily behaviors.

Moreover, this study relies on the CDRs as the internet signaling data were not available in 2018. Nevertheless, the CDRs includes active confirmation data per hour, event data of phone messages, and location area code (LAC) switching data beyond a certain range. These three main sources of CDRs can provide 30–50 LBS records per person per day, which is enough for this study.

In total, around 1012 million cellular data records generated by 20 million users in two weeks were provided. The recording intervals of were not fixed but within 30 min. As shown in Figure 3, the medium service area of cellular base stations within historical conservation subsets in central Shanghai was 0.03 km², while the medium area of historical conservation subsets is 1.04 km², which enabled an acceptable estimating precision of users' location. The Voronoi diagram in Figure 4 indicates the fine-grained distribution of cellular base stations in central Shanghai.

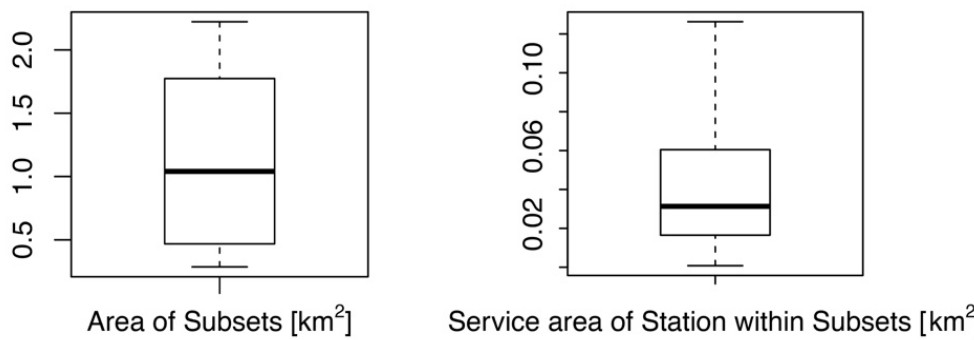

**Figure 3.** Comparison between area of subsets and service area of station within subsets.

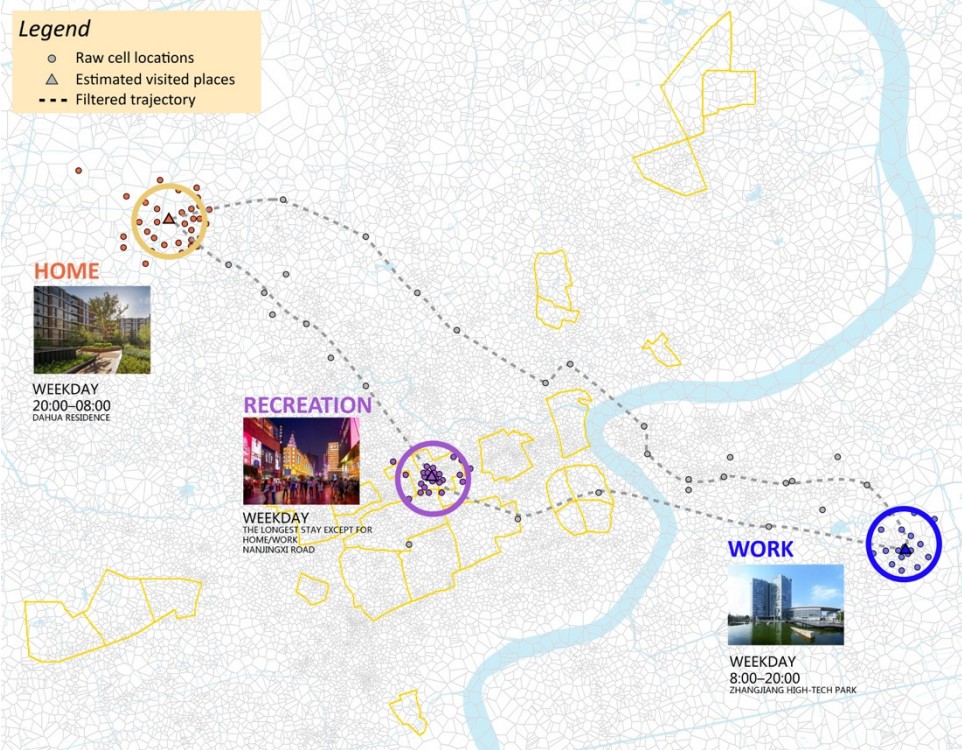

**Figure 4.** The constructed trips and visited places of an anonymous local resident. Dots represent the raw cell locations and the dash line is the filtered trajectory. Big circle is the range of clustering of dots, which means the staying duration of this position is long.

Anonymous IDs allow recognition of data points generated by the same tourist [30], while the trajectory generated in a certain period of time is aggregated through the linkage of discrete personal roaming data point. Different groups of users were classified via positioning points following the criteria below:

(1)  First, we classified the local and non-local IDs at the scale of the whole Shanghai city. Those IDs which consistently stayed in the Shanghai at night were regarded as residents of Shanghai. IDs that travelled for outside to Shanghai were considered to be non-local visitors.

(2)  Then, we identified the residential, working, and recreational places for each resident of Shanghai. The location where one ID stayed between 8:00 p.m. and 8:00 a.m. over a period of more than seven days was marked as the "home" of this ID. Correspondingly, where one ID stayed between 8:00 a.m.–8:00 p.m. for over seven days was identified as the place where this ID works. Beyond the residential and working place, the rest place where one ID spent most time was identified as the day's place of "recreation."

Figure 4 shows a series of trips and places visited for one local ID. Nanjingxi Road Historical Area could be identified as his or her place of recreation on one weekday.

(3) Based on that, the four kinds of people related to the 33 historical conservation sites, i.e., local residents, local commuters, local tourists, and non-local tourists, were identified. Specifically, the local residents and local commuters are the IDs obtaining residential and working places in these historical conservation sites, respectively. The local tourists are the IDs obtaining recreational places in the sites. The non-local tourists are non-local visitors spending at least three hours in the sites.

(4) With the help of ArcGIS, we were able to identify the number of people within 33 historical conservation sites. Specifically, the numbers of local residents, local commuters, local tourist, and non-local tourists were derived for each site. According to the proportion of residents, commuters, and tourists, the degree of balance of urban and tourism development among these historical conservation areas can be measured.

*3.4. Visualizing the Degree of Balance among Different Historical Conservation Areas*

The 'balanced' urban development here represents the situation historical conservation areas obtaining an equilibrium among the three groups of people, i.e., local residents, commuters, and tourists. Specifically, the identification of these three groups is achieved based on cellular data. The ternary graph (Figure 5) is an analytical tool to visualize the 'balance' degree and express three groups of people as proportions. It is a barycentric plot on three variables which sum up to a constant, which is a widely used tool to show the composition of systems with three indicators [38]. In a ternary plot, the values of the three variables a, b, and c must sum up to a constant. Usually, this constant is represented as 1.0 or 100%. Because the three numerical values cannot vary independently—there are only two degrees of freedom—it is possible to graph the combinations of all three variables in only two dimensions. In recent years, some urban researchers have adopted the ternary graph as a tool to test equilibrium among three urban functions [39].

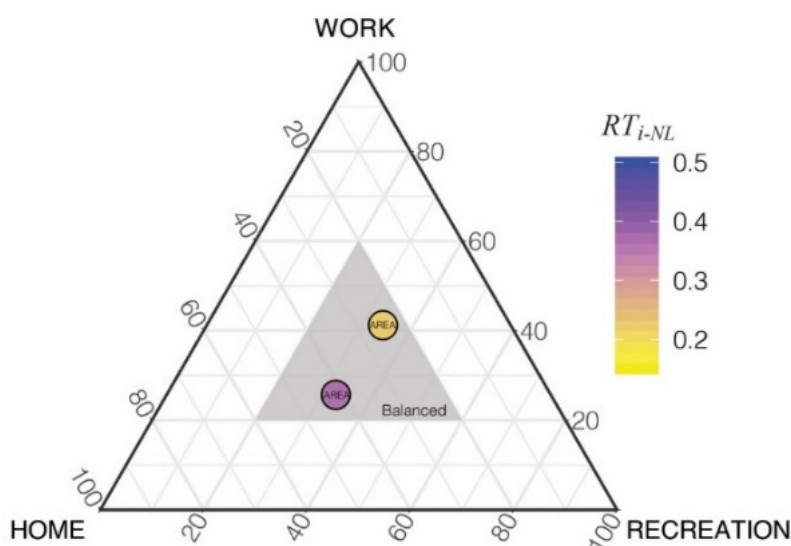

**Figure 5.** Ternary graph for visualizing the balanced urban and tourism development area; $RT_{i-NL}$ represents the ratio (range: 0–1) of non-local tourists.

In this study, the proportion of three groups of people ($RH_i$, $RT_i$, $RW_i$) were calculated based on the above cellular data processing results. As shown on Figure 4, the sides of "HOME," "RECREATION," and "WORK" are represented on the of $RH_i$, $RT_i$, $RW_i$, scales, respectively, with each angle point representing the 100% ratio of "HOME," "RECRE-ATION," and "WORK." These data points are plotted according to the calculated data.

The threshold ratio defining the balanced degree of historical subsets in this ternary graph is based on the concept of tourist to resident ratio. This ratio indicates travel intensity,

with the threshold for balanced tourism development depending on residents' attitude towards tourism [6]. Li, Xie, and Wang's [40] empirical study shows that historical areas with a ratio of tourist to local resident between 1.0 and 1.5 can be regarded as "balanced," where tourists and local residents can benefit from each other. Areas with a ratio above 1.5 tend to be tourism-oriented neighborhoods. Moreover, there is also some research on the job to housing ratio. Studies find that a job to housing ratio ranging from 0.75 to 1.25 leads to a "balanced" community [41,42]. Based on this, we propose a hypothetical ratio of 1:1:1 (33.3%:33.3%:33.3% in the ternary graph) to be the ideal balanced point for three groups of people. The balanced area based on this ideal point can be visualized as the grey hatched triangle with the buffer zone. In other words, the points within the grey hatched triangle close to the center are identified as balanced developed subsets.

The points located on the periphery of the graph indicate less balanced or unbalanced development areas. In addition, the color of data points indicates the ratio non-local tourists occupied among all the tourists in this area ($RT_{i-NL}$). Blue represents a higher ratio of non-local tourists, while yellow represents a lower ratio.

Moreover, areas were labeled as "always unbalanced" if the points are never located within the hatched grey area during weekdays and weekends. Areas were labeled as "sometimes balanced" if they are located in the hatched area for either weekdays or weekends. The historical conservation area which was plotted within the balanced triangle for both weekdays and weekends was classified as being "always balanced."

*3.5. Detecting Contributing Features to the "Degree of Balance" through Statistical Analysis*

A multinomial logistic regression was constructed to explore the relationship between related urban attributes and degree of balance mapped via the ternary graph. This statistical model was based on the Hedonic price model [43]. Specifically, the degrees of balance are typically regressed against a large set of predictor variables with external factors including urban scale features and internal factors including local facilities and functions based on PoIs and built environment features.

3.5.1. Independent Variables Related to Urban Development

Independent variables were selected according to elements related to urban management (Table 2). Detailed distribution of these variables can be checked in Figure A1 in Appendix A. First, urban scale features include distance to the city center (DISC) and Huangpu River (DISR). Renmin Square is considered a symbol of city center here. Second, local urban facilities and land-use functions serving both tourism and residences were included as well. Public transportation is an economical way to enable tourists to move around between different destinations. As mentioned by Le-Klaehn and Hall [44], most non-local tourists depend on public transportation, and tourist destinations with effective and accessible public transport networks have a higher possibility of attracting more attention from tourists. Therefore, the number of bus stations (BUS) and metro stations (MET) per square meters for each target subsets are included in the model. Moreover, hotels (HOT), which are an important factor in boosting local tourism, have been added as well. In addition, commercial space (COM), public service (PBS), and catering space (CTS) were included, as they can also promote tourism by integrating tourists with local residents through socializing and meeting in this space. These supplied services including urban infrastructure or amenities determine the lowest realized tourism levels [9].

**Table 2.** Predictor variables and descriptive characteristics for logistic regression.

| Predictor Variable | Variable Name | Units | Mean | S. D | Min | Max |
|---|---|---|---|---|---|---|
| Urban Scale Features | | | | | | |
| Distance to City Center | DISC | m | 8807.1 | 8464.6 | 123.8 | 33,172.9 |
| Distance to Huangpu River | DISR | m | 6315.2 | 6123.9 | 437.2 | 23,602.8 |
| Local Facilities and Functions | | | | | | |
| Number of Metro Station | MET | / | 1.7 | 1.7 | 0.0 | 6.0 |
| Number of Bus Station | BUS | / | 28.3 | 18.5 | 1.0 | 80.0 |
| Number of Commercial | COM | / | 9.2 | 8.1 | 0.0 | 28.0 |
| Number of Public Service | PBS | / | 8.7 | 9.2 | 0.0 | 32.0 |
| Number of Hotel | HOT | / | 21.4 | 16.1 | 0.0 | 67.0 |
| Number of Catering Service | CTS | / | 378.5 | 327.1 | 1.0 | 1230.0 |
| Built Environment Features | | | | | | |
| Area of the District | km$^2$ | km$^2$ | 1.1 | 0.7 | 0.3 | 2.2 |
| Area of the Buildings | ABUI | km$^2$ | 1.3 | 0.9 | 1.5 | 3.8 |
| Mean Height | MHEI | m | 5.0 | 1.5 | 2.3 | 9.6 |
| Medium of Urban Block Area | MEUBA | m$^2$ | 236.4 | 64.7 | 120.6 | 353.3 |
| Accessibility | ACC | / | 147,624,518.6 | 113,685,533.3 | 9,688,160.2 | 443,784,990.3 |

However, some other built environment features related to urban planning management are included in this model as well, such as mean height (MH), area of the district (ADIS)/buildings (ABUI), and medium of urban block area (MEUAB). Street accessibility measured via space syntax was also used as the last predictor variable in this MLR models. Space syntax mainly focuses on detecting how spatial configurations affect behaviors within the urban environment. The measurement of choice is defined as the number of least-angle-change paths between all of the other links that pass-through a given segment [45]:

$$C_b(p_i) = \sum_{j=1}^{n} \sum_{k=1}^{n} g_{ik}(p_i) / g_{jk} \ (j < k)$$

where $g_{ik}(p_i)$ = the number of shortest paths between node $p_j$ and $p_k$ which contain $p_i$, $g_{jk}$ = the number of all shortest path between node $p_j$ and $p_k$. It reflects the potentials of travel for pedestrians or drivers [46]. Specifically, the choice value calculated by the sDNA software represents the accessibility of individual streets. Conversion from streets to historical conservation subsets was then achieved through buffer analysis in GIS.

### 3.5.2. Multinomial Logistic Regression (MLR)

The multinomial logistic regression model is estimated in RStudio using the "nnet" package [47]. $P_{i1}$, $P_{i2}$, $P_{i3}$ are the probability associated with the choice of the three "degrees of balance" respectively by $i$th individual areas, which is represented as:

$$P_{ij} = \frac{\exp(x_i' \beta_j)}{1 + \sum_{k=2}^{3} \exp(x_k' \beta_j)} P_{i1} + P_{i2} + P_{i3} = 1$$

Here, $x_i'$ is the $i$th row of the model $X$, while $\beta_j$ is a vector of all the regression coefficient for each $j$ ($j$ = 1, 2, 3) categorical labels. $P_{i1}$ (label: always unbalanced) is the baseline of this MLR model. Essentially, MLR could form two logits and be transformed into linear regression problem as fitting the two log odds:

$$Logit1 = log \frac{P_{i2}}{P_{i1}} = x_i' \beta_2 \, Logit2 = log \frac{P_{i3}}{P_{i1}} = x_i' \beta_3$$

Namely, the proposed model could also be interpreted as the following form:

The log odds of two "degrees of balance" = $f$(DISC, DISR, MET, BUS, COM, PBS, ... , ACC).

## 4. Results

*4.1. Ternary Graphs as a Tool to Classify the Balanced Degrees for Urban Historical Conservation Areas*

From Figure 6A,B, the points in the ternary graph show a movement trend towards the angle of "RECREATION" from weekdays to weekends, while the three points of QB, SY1, and HQ3 demonstrate the most significant movement trend. QB (Qibao Watertown), SY1, HQ3 (*Shanghai Zoo*) have the largest movement from Figure 6A,B, suggesting that these three have the largest increasing rate of tourists. However, TL1, YY1, and NJ2 remain almost the same from Figure 6A,B).

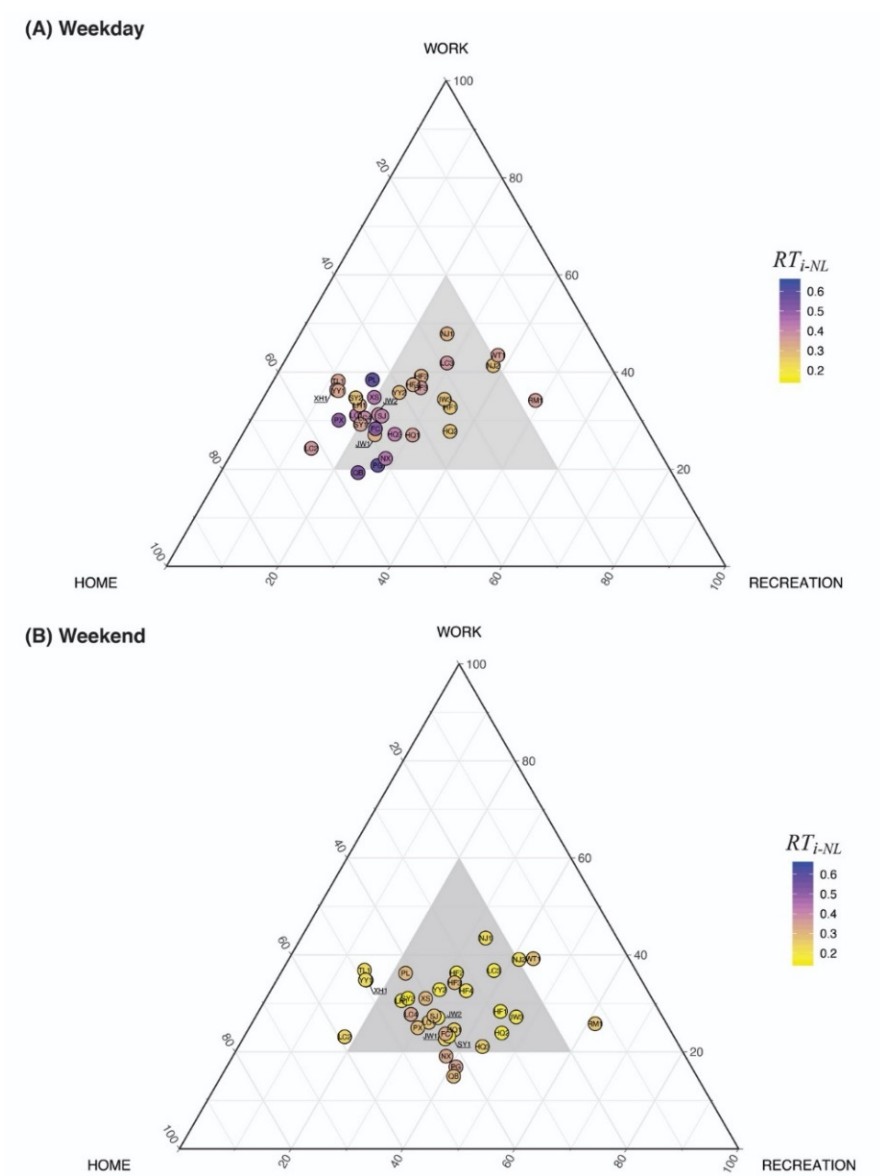

**Figure 6.** (**A**) The ternary graph of 33 subsets on weekdays. (**B**) The ternary graph of 33 subsets on weekend ($RT_{i - NL}$) represents the ratio of non-local tourists among all the tourists.

High $RT_{i-NL}$ indicates that PL, PG, and QB have significantly higher proportion of non-local tourism than the others. WT1 is near the Huangpu River and has famous waterfront skyline scenery. LC1 is symbolic for YU Garden, Chenghuang Temple. Both

WT1 and LC1 are famous tourism destinations of Shanghai. $RT_{i-NL}$ of all the historical conservation areas decreased from weekdays to weekends, likely due to the rise of local tourists on weekends. HF1, HQ2, and SY2 are destinations preferred by locals rather than non-local tourists.

On weekdays, LC1, LC2, LH1, RM1, SY2, TL1, WT1, XH1, YY1, PX, QB, PL, and XS are biased instead of being balanced-developed subsets. On weekends, LC2, NJ2, RM1, WT1, XH1, YY1, PG, QB, and NX are out of the boundary of the balanced triangle. Therefore, in Table 3, LC2, RM1, WT1, YY1, and QB are the five out of 33 subsets labeled as "always unbalanced"; LC21 LH1, NJ2, SY2, TL1, PG, PX, PL, NX, and XS are labeled as "sometimes balanced," while the majority of these subsets including the whole HF (HF1, 2, 3, 4), the whole HQ (HQ1, 2, 3), the whole JW (JW1, 2, 3),), LC3, NJ1, SY1, YY2, SJ, and FC are "always balanced."

**Table 3.** Voting on MLR classification labels of dependent variables.

| Area | HF1 | HF2 | HF3 | HF4 | HQ1 | HQ2 | HQ3 | JW1 | JW2 | JW3 | LC1 | LC2 | LC3 | LC4 |
|---|---|---|---|---|---|---|---|---|---|---|---|---|---|---|
| Weekday | B | B | B | B | B | B | B | B | B | B | B | - | B | B |
| Weekend | B | B | B | B | B | B | B | B | B | B | B | - | B | B |
| Final Label | AB | AB | AB | AB | AB | AB | AB | AB | AB | AB | BS | AU | AB | AB |
| | LH1 | NJ1 | NJ2 | RM1 | SY1 | SY2 | TL1 | WT1 | XH1 | YY1 | YY2 | PG | PX | QB |
| Weekday | - | B | B | - | B | - | - | - | - | - | B | B | - | - |
| Weekend | B | B | - | - | B | B | B | - | - | - | B | - | B | - |
| Final Label | BS | AB | BS | AU | AB | BS | BS | AU | AU | AU | AB | BS | BS | AU |
| | PL | NX | SJ | XS | FC | | | | | | | | | |
| Weekday | - | B | B | - | B | | | | | | | | | |
| Weekend | B | - | B | B | B | | | | | | | | | |
| Final Label | BS | BS | AB | BS | AB | | | | | | | | | |

"AU": Always Unbalanced; "BS": Balanced sometimes; "AB": Always Balanced.

As shown in Figure 7, the development of HF, HQ, and JW is relatively homogenous from the clustered pattern in the spatial distribution of balanced degree. However, LC has quite significant divergence. LC1 and LC2 are residential-biased while LC3 and LC4 are balanced comparably.

*4.2. Multinomial Logistic Regression (MLR) Classification*

A multinomial logistic regression model was used to assess the marginal effect of urban features, i.e., urban scale features, local facilities and functions, and built environment features, on the likelihood of categories to be balanced. In Figure 8, due to the high correlation between ratio of local tourists of weekdays and weekends, distance to city center and Yangtze River, area of buildings, number of public services, catering services, and hotels, therefore, $RT_{i-NL}$ of weekdays, DISR, ABUI, HOT, CTS are removed to acquire the highest accuracy of prediction from the models.

All other urban features with statistical significance have small standard errors, indicating that these urban features play an important role in affecting the degree of balanced development. As discussed in Section 3.5.2, the coefficients in the table are for estimating log odds ($\beta_2$ for $log\frac{P_{i2}}{P_{i1}}$, $\beta_3$ for $log\frac{P_{i3}}{P_{i1}}$, $\beta_3 - \beta_2$ for $log\frac{P_{i3}}{P_{i2}}$). Then, the exponentiation of the coefficient represents the odds ratio showing the constant effect of the specific predictor variable. If the exponential value is higher than one, the likelihood of choosing the category of nominator other than denominator of the odds will increase.

In Table 4, $exp(\beta_2)$ and $exp(\beta_3)$ are compared to identify variables that effectively distinguish "Always Balanced" subsets from the others ("Balanced Sometimes" and "Always Unbalanced"). The exponential value of DISC and ACC are around 1, which indicates that predictor variables like distance to city center (DISC) and local accessibility (ACC)

have less influence on the classification. This also suggests that features on the urban scale have little impact on local balanced urban development. The count of bus stations (BUS), the total area of subsets (ADIS) and medium of urban block area (MEUBA) have negative coefficients with these two logits. With more bus stations and larger area of subsets, the probability of the subsets being always balanced increases. The count of metro stations (MET), the commercial spaces (COM), the count of public services (PBS), and the ratio of non-local tourists ($RT_{i-NL}$) have both exponential value higher than one. With more metro stations, commercial spaces, public services, and more non-local tourists, the probability of the subsets being always balanced decreases. The mean height (MHEI) has a different effect with the two exponential values, making it difficult to draw inferences.

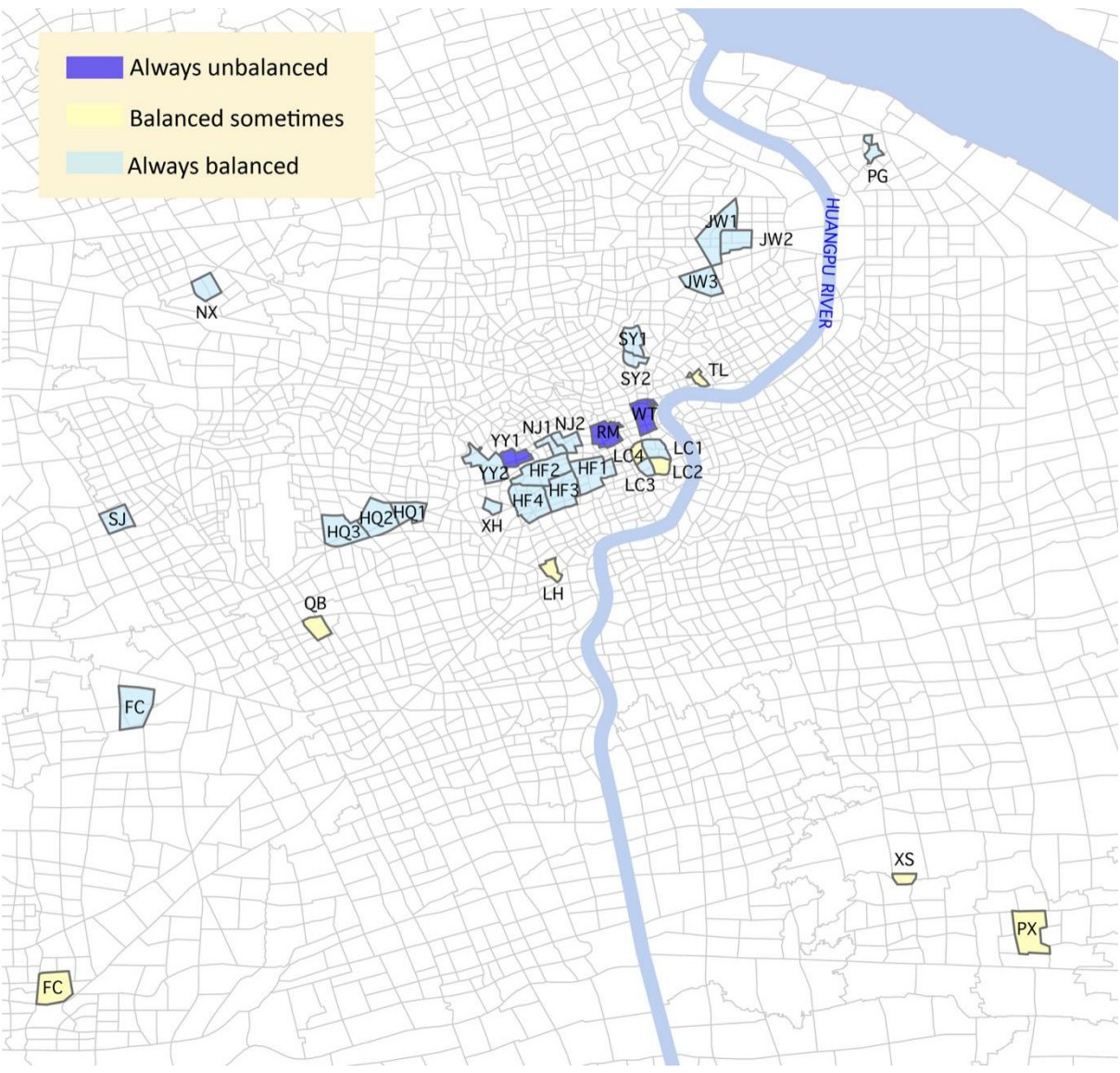

**Figure 7.** The spatial distribution of balanced degree related to different historical conservation areas.

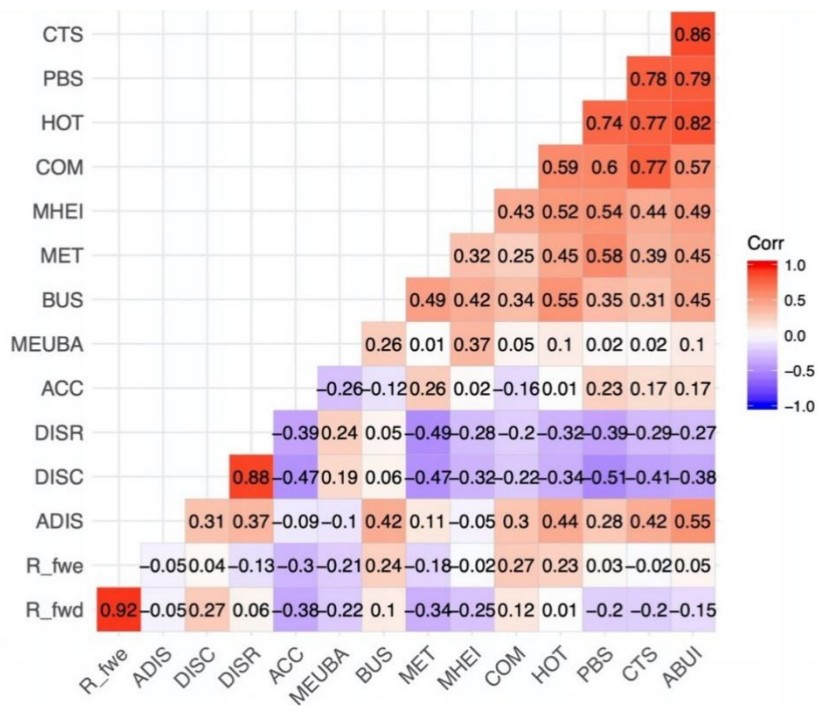

**Figure 8.** Correlation Matrix of Variables.

**Table 4.** The summary of MLR results.

| Predictor Variables | Balanced Sometimes/Always Balanced | | | Always Unbalanced/Always Balanced | | |
|---|---|---|---|---|---|---|
| | Coefficient ($\beta_2$) | Standard Error | $exp$ ($\beta_2$) | Coefficient ($\beta_3$) | Standard Error | $exp$ ($\beta_3$) |
| Intercept | 0.1836 *** | $1.1891 \times 10^{-17}$ | 1.2015 | 0.2478 | $1.4485 \times 10^{-17}$ | 1.2812 |
| **Urban Scale Features** | | | | | | |
| DISC | 0.0005 *** | $7.7224 \times 10^{-14}$ | 1.0005 | 0.0003 *** | $6.0957 \times 10^{-14}$ | 1.0002 |
| **Local Facilities and Functions** | | | | | | |
| BUS | −0.0493 *** | $2.3312 \times 10^{-16}$ | 0.9518 | −0.0059 *** | $3.5163 \times 10^{-16}$ | 0.9941 |
| MET | 0.0757 *** | $2.2123 \times 10^{-17}$ | 1.0786 | 0.3052 *** | $2.7608 \times 10^{-17}$ | 1.3569 |
| COM | 0.2293 *** | $8.8610 \times 10^{-17}$ | 1.2577 | 0.0937 *** | $8.1426 \times 10^{-17}$ | 1.0982 |
| PBS | 0.2494 *** | $1.2127 \times 10^{-16}$ | 1.2833 | 0.0679 *** | $7.9312 \times 10^{-17}$ | 1.0702 |
| **Built Environment Features** | | | | | | |
| ADIS | −0.0782 *** | $1.0784 \times 10^{-15}$ | 0.9340 | −0.0635 *** | $8.2946 \times 10^{-16}$ | 0.9384 |
| MHEI | −0.1866 *** | $5.9841 \times 10^{-17}$ | 0.8298 | 0.7114 *** | $7.7382 \times 10^{-17}$ | 2.0368 |
| MEUBA | −0.0077 *** | $2.6637 \times 10^{-15}$ | 0.9923 | −0.0087 *** | $3.3376 \times 10^{-15}$ | 0.9913 |
| ACC | $5.3720 \times 10^{-9}$ | $3.1679 \times 10^{-9}$ | 1.0000 | $-3.9948 \times 10^{-9}$ | $3.0882 \times 10^{-9}$ | 1.0000 |
| **Non-local tourists' ratio ($RT_{i-NL}$)** | | | | | | |
| weekend | 0.0061 *** | $2.6775 \times 10^{-18}$ | 1.0061 | 0.0711 *** | $3.2859 \times 10^{-18}$ | 1.0737 |

Wald 'z test for significance; *** Significant at the 0.001 level.

The odds ratio of $exp(\beta_3 - \beta_2)$ in Table 5 with $exp(\beta_3)$ in Table 4 are compared to identify variables that effectively distinguish "Always Unbalanced" subsets from the others ("Balanced Sometimes" and "Always Balanced"). The number of metro stations (MET), mean height (MHEI), and the ratio of non-local tourists ($RT_{i-NL}$) have both exponential values higher than one, indicating that areas with increasing number of metro stations, higher mean height, and higher ratio of non-local tourists on weekends tend to be unbalanced all the time. The medium of urban block area (MEUBA) shows less than one in both exponential values, which indicates that subsets with large urban block areas have a less likelihood of being unbalanced all the time. The count of bus stations (BUS), the commercial spaces (COM), the count of public services (PBS), and total area (ADIS) shows

adverse effects with the two exponential values, which cannot be sufficient variables to differentiate levels here.

**Table 5.** The calculation results of "Always Unbalanced" versus "Balanced Sometimes".

| Always Unbalanced/Balanced Sometimes | (Intercept) | DISC | BUS | MET | COM | PBS |
|---|---|---|---|---|---|---|
| $exp(\beta_3 - \beta_2)$ | 1.0664 | 0.9997 | 1.0444 | 1.2580 | 0.8732 | 0.8340 |
| | ADIS | MHEI | MEUBA | ACC | Weekend | |
| $exp(\beta_3 - \beta_2)$ | 1.0047 | 2.4547 | 0.9990 | 1.0000 | 1.0672 | |

Modeling classification performance was evaluated via a confusion matrix. In the following Table 6, a confusion matrix is presented for all classified reviews in three degrees of balance according to the above MLR models. The value of accuracy representing the proportion of subsets correctly classified divided by total observations is 81.8%, which indicates the model produces accurate results especially for identifying the always balanced subsets.

**Table 6.** The confusion matrix for accuracy assessment.

| Reviews | Classified as AB | Classified as BS | Classified as AU |
|---|---|---|---|
| AB | 17 | 1 | 0 |
| BS | 1 | 8 | 2 |
| AU | 0 | 2 | 2 |

## 5. Discussion

### 5.1. A New Perspective for Measuring the Degree of Balance between Urban Development and Tourism Planning

The issue of balanced tourism development is capturing scholarly interest due to the increasing number of cities facing conflict between tourism and local community development. Evaluating the current degree of balance is always a difficult task requiring complex data and analyses. Therefore, empirical analyses in this direction are relatively rare [12,13].

An analytical approach was employed, utilizing ternary graphs to develop a quantitative-based qualitative classification to visualize the degree of balance, which was an intangible issue in the past. As stated by UNWTO [6], balanced tourism and urban development should never be limited to tourism related industry. It is important to involve related stakeholders to ensure local residents have a thorough understanding of the positive effects of tourism. Based on a review of existing literature, this study extends previous analyses focusing on the tourist-resident ratio by adding commuters into the consideration to construct a ternary graph. This new approach is able to demonstrate the balanced condition of three stakeholders within the local community: the tourists, commuters, and residents.

In general, the ternary graph approach proposes a new perspective to monitor the degree of balance. This approach might be helpful for further strategies to protect unique local characteristics while stimulating cultural tourism [48]. The successful application in Shanghai historical conservation areas shows its potential to be applied in many other historical cities facing the same demand of keeping a balance between urban and tourism development. Insights achieved from this tangible and continuous analytical approach may assist efficient policymaking in both urban development and tourism planning.

### 5.2. Urban Management Implications towards Balanced Development for Historical Conservation Area

First, related management strategies promoting balanced urban development should not be simplified as the restraining of tourism industry near historical conservation areas. In turn, it is important to strengthen local-scale infrastructures which can be easily accessed

and used by residents nearby. As mentioned in our study, a large number of metro stations contributes to a business-biased area and an unbalanced area, which agrees with existing empirical studies [49]. In turn, bus stations tend to show positive effects on balanced development. This might be because metro stations are usually integrated with business-dominant development areas, encouraging the large amount of tourist inflow [50]. However, bus stations are usually used by local residents and commuters rather than tourists. Therefore, appropriate urban management bringing benefits and accessibility for local residents might play an important role in promoting balanced urban development for historical conservation areas.

Existing urban management strategy often tends to remove all commercial facilities from the historical convention sites as intensive commercial development does bring negative effects. Our study confirms that too many commercial and public services can bring negative effects on balanced development of local communities. Nevertheless, there is no evidence that integrating any commercial and services will result in an always unbalanced area for the historical conservation area. It suggested that integrating moderate public and commercial services with historical conservation areas could encourage communication between residents and visitors and support all the stakeholders in the neighborhood, making this a desirable strategy for urban development.

In terms of built environment features, building height might lead to unbalanced local communities. Therefore, building height should be restricted, which matches with the consensus in urban planning [51]. In addition, our empirical study reveals that larger block size and area of subsets may promote balanced tourism development, which is contrary to the advocacy of smaller block areas and denser street networks [52]. This situation can be explained as slightly larger block areas indicating less pedestrian movement and commercial potential can keep residential life within the block. Therefore, relatively larger street blocks should be encouraged in the following urban management implications to avoid over-tourism in historical areas.

### 5.3. Large-Scale Analysis with a Human-Perspective: Advantages Based on Cellular Data

This study also revealed advantages of distinguishing people's spatio-temporal patterns based on a large-scale analysis of cellular data. Cellular data could be collected for millions of participants within urban regions. Meanwhile, details of individual behavior based on tracing routes were identified with an acceptable spatio-temporal resolution. In addition, the combination of spatial and temporal distribution provides an approach to draw personal profits of cell phone users. We are able to distinguish tourists, commuters, and local residents from millions of cell phone users with high accuracy. Compared with GPS tracking using a limited amount of GPS trackers [53,54], using a large-scale analysis with a human-perspective provides new research potential for identifying the tourists' behavior.

As stated by Brown [55], tourism development activity was always a responsive decision process instead of a preplanning outcome. This new analytical framework based on cellular data is capable of measuring the intangible balanced development and assisting in a more evidence-based and scientific-oriented decision support for urban management of historical conservation area.

### 5.4. Limitations and Future Steps

First, the situation that one individual owns multiple mobile phones does exist and is difficult to avoid. Due to the privacy protection requirements, all user IDs are encrypted and anonymous in the original data, and personal information cannot be obtained. Nevertheless, empirical studies show that the users holding more than one mobile phone are a minority and thus would not affect the result too much. Second, although China Mobile obtains the highest market share in Shanghai, current analyses that compare the number of mobile devices to the target population still pose some challenges and uncertainties. A combined verification of cell phone data from all three mobile communication companies

with other Internet LBS data might help to address these two limitations in the future. Third, due to technical restriction, spatial resolution of cell phone positioning is relatively limited. Adding cell phone GPS records could be a supplement to provide higher accuracy in future studies. In addition, the "balanced" area identified in a ternary graph is based on a hypothetical ratio generated from previous literature, which has some space to be refined. An integrated study could be made in future studies to provide more concrete information in this direction. Moreover, the two-week time span is relatively short. A longer observation covering at least one whole year would help understand the effect of seasons in our next study.

In addition, current data-driven approaches which rely on large amounts of cellular data cannot reveal people's motivation and detailed behavior effects. Instead, we firstly classify the people into groups, i.e., residents, commuters, and tourists following the selection principles. Based on the hypothesis that the people within one group will share more similar behavioral patterns in local communities, we then evaluate their impact on balance degree of local communities. The similar analytical pattern has been widely applied in a series of studies [31,56–58]. Although it cannot provide detailed behavior records for everyone, this data-driven approach is still able to reveal a big picture with the help of big data.

Nevertheless, it is a pity that personal motivation and behavioral effects were not involved in current analysis. Classical methods, e.g., public survey and in-depth interview, can address this issue well but they are time-costly and can only be applied within a small scale. Large-scale cellular analysis is capable of covering a whole city but would miss the concern of personal motivation and behavioral effects. We hope future technical innovations would help to achieve the co-present of personal motivation, behavioral effects, and spatial-temporal distribution, and finally promote a more comprehensive analysis.

It is also worth mentioning that current definitions and measurements of the balance degree between urban and tourism development are from previous studies. The changing of balancing standards may cause different results. Current standards applied in Shanghai might not perform well in other cities, as cities among different cultures and regions may obtain different criteria. Nevertheless, this limitation might not affect the key contribution of our study. As we claimed in the abstract, the focus of this study is to explore a methodological approach for measuring the degree of balance between urban and tourism development. This analytical approach based on cellular data works well, as shown in the current paper. However, it is important for us to consider the setting of standard when we apply it in various kinds of cases.

## 6. Conclusions

The main contribution of this research is creating an analytical framework using ternary graphs to evaluate the degree of balance between urban development of an area with tourism resources, which was previously hard to visualize and measure. This paper has also identified that some noteworthy features impact balanced development of local communities. An understanding of these urban features will assist efficient decision-making to promote balanced urban development.

The framework is not targeted at restraining the development of tourism for the historical conservation area, but at keeping all the invested infrastructure balanced and fully functional. Balanced tourism has been shown to not only promote local economyeconomies and offer employment opportunities, but also correlate closely with socio-economic development of cities and the convenience of life of local residents. Moreover, some urban planning strategies need to be modified to foster more balanced tourism.

Using cellular data in urban management brings a new perspective to measuring the degree of development balance for historical conservation areas. Future research should build a more comprehensive user portrait with combined application of fine-grained cellular data and multi-sourced urban data to assist data-informed urban management.

**Author Contributions:** Conceptualization, C.S. and Y.Y.; methodology, Y.Y.; software, M.L.; valida-
tion, M.L., Y.Y. and C.S.; formal analysis, M.L.; investigation, M.L.; resources, C.S.; data curation,
C.S.; writing—original draft preparation, M.L.; writing—review and editing, Y.Y.; visualization, M.L.;
supervision, Y.Y.; project administration, Y.Y.; funding acquisition, Y.Y. All authors have read and
agreed to the published version of the manuscript.

**Funding:** This research was funded by the National Natural Science Foundation of China, grant num-
ber 52078343 and 51808392, and Natural Science Foundation of Shanghai, grant number 20ZR1462200.

**Institutional Review Board Statement:** Not applicable.

**Informed Consent Statement:** Not applicable.

**Acknowledgments:** We would like to express our appreciation for the editor and two anonymous
reviewers for the time and effort they have taken to provide insightful guidance. We also appreciate
the support from three research assistants, Rong HUANG, Gong ZHANG, and Dan QIANG.

**Conflicts of Interest:** The authors declare no conflict of interest.

**Appendix A**

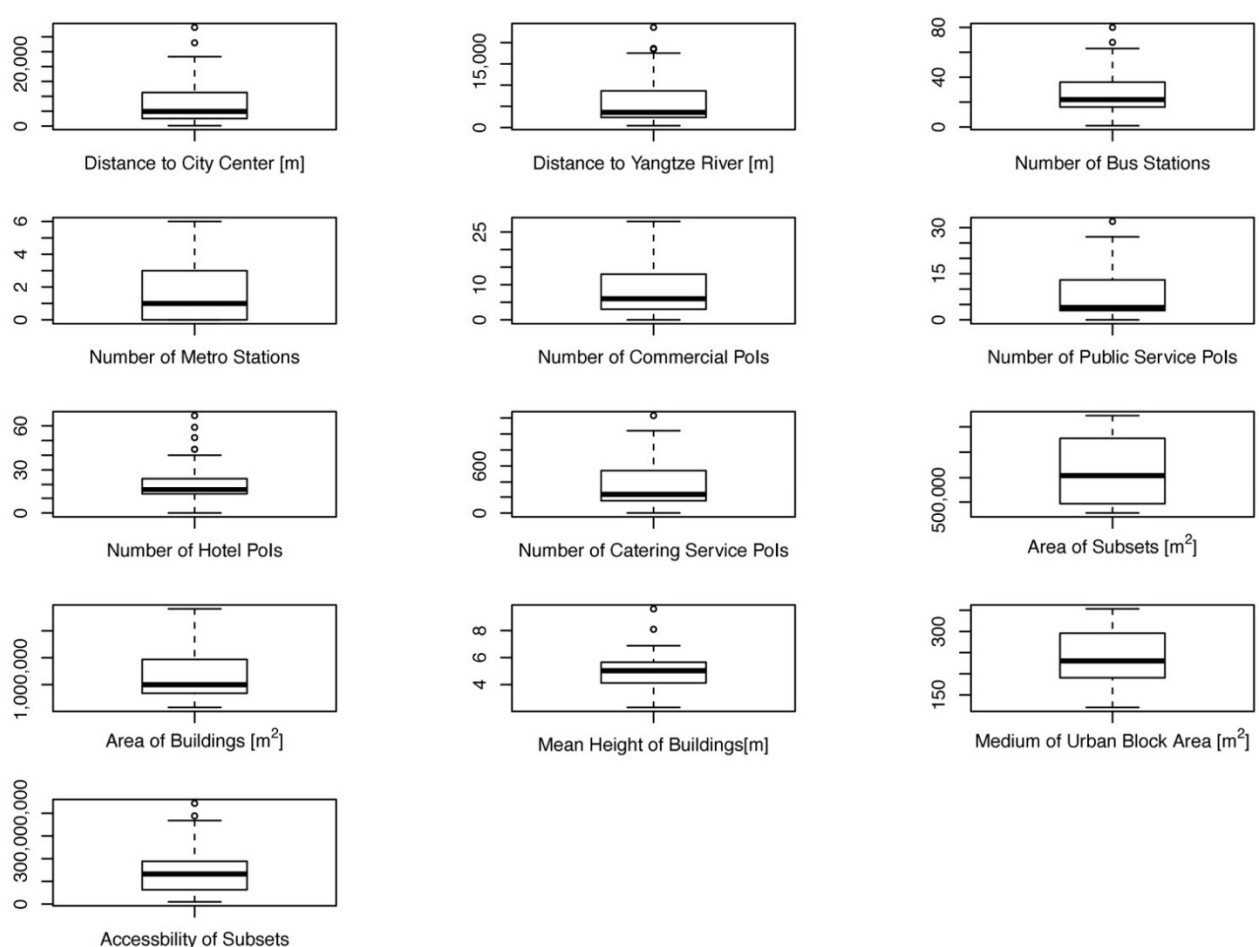

**Figure A1.** Data description of independent variables.

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
