# Peer review of "Measuring the Degree of Balance between Urban and Tourism Development: An Analytical Approach Using Cellular Data"

_sustainability, doi:10.3390/su13179598_

Round 1
Reviewer 1 Report
Dear Authors: it is an interesting and a very relevant paper. The methodological framework that you apply to measure the pressures tourism exerts on sites of historical value is a very interesting one. To improve the value of the paper, I suggest that you:
- rephrase the key research objective: at present it is nor very clear whether you want to focus on tourism only, or whether perhaps on tourism in connection to urban development. If the latter is the case, then you may want to be more precise in defining the objective of your work.
- In the same vein, since there is essentially no discussion section, where you would explain the reader what the elaborate measurements stand for, it is difficult to discern what the value added of the paper is and which ways your findings may be applicable in other contexts.
Author Response
Reviewer 1
Dear Authors: it is an interesting and a very relevant paper. The methodological framework that you apply to measure the pressures tourism exerts on sites of historical value is a very interesting one. To improve the value of the paper, I suggest that you:
1) rephrase the key research objective: at present it is not very clear whether you want to focus on tourism only, or whether perhaps on tourism in connection to urban development. If the latter is the case, then you may want to be more precise in defining the objective of your work.
Response: Many thanks for your kind reminder. This study focuses on urban development for the historical conservation area which have high possibility to be over-tourism-oriented. Thus, we are attempting to develop an analytical approach with the help of multi-sourced urban data for enabling urban planners and tourism managers to evaluate the degree of “balance” especially for area with tourism resources. Achievements in this direction help to provide a more evidence-based decision-making process towards balanced urban development.
Revision: We have highlighted our research objective in Line 52-60 and research questions in Line 117-123. We have also revised the “informing better decision” part of Figure 1 to clarify our research focus (Line 125)
2) In the same vein, since there is essentially no discussion section, where you would explain the reader what the elaborate measurements stand for, it is difficult to discern what the value added of the paper is and which ways your findings may be applicable in other contexts.
Response: Many thanks for this helpful comment. Revisions have been made in our discussion to highlight the contributions of this study. First of all, it provides a new perspective for measuring the degree of balance between urban development and tourism planning (Line 414-415). Second, a series of urban management implications towards balanced development for historical conservation area have been developed (Line 438-439). Moreover, it is also an exploration to achieve large-scale spatial-temporal analysis with a human-perspective via cellular data (Line 470).

Reviewer 2 Report
The article is interesting and innovative and tackles the interesting issue of how big data paradigms can be used to understand the behavior of tourists and day visitors better and use some of this information to improve (tourism) policies and make tourism more sustainable. It has a number of important yet not fatal shortcomings the authors should pay some attention to.
First of all, the concept of 'balance', central in the paper is not very well explained nor sufficiently well supported by literature. Is balance just a question of how much time somebody spends in a place per day, or does it have to do with what people exactly are doing there, including spending money or making noise, polluting and so forth. Obviously, with Telco data not much can be said about all this.
Secondly, the definition of tourists and day trippers, also in literature, is not only dependent on time and place but also on motivation. Again, and this is also related with the first point, the interpretations of Telco data are virtually useless when authors take shortcuts when discussing definitions.
Obviously, the whole discussion the authors develop throughout the paper suffers from these two crucial shortcoming. There is, however, plenty of literature available to address these two important issues and I trust the authors will be able to adjust the paper using this. The paper is interesting and innovative enough to spend some more energy in perfectioning it.
Author Response
First, we highly appreciate the editor and the two anonymous reviewers taking the time to offer us comments and insights related to the paper. We greatly appreciate the opportunity to extend our thinking and to improve our manuscript. The responses are given according to the sequence of the reviewer’s comments. All revisions have been highlighted as blue in the manuscript.
Reviewer 2
The article is interesting and innovative and tackles the interesting issue of how big data paradigms can be used to understand the behavior of tourists and day visitors better and use some of this information to improve (tourism) policies and make tourism more sustainable. It has a number of important yet not fatal shortcomings the authors should pay some attention to.
1) First of all, the concept of 'balance', central in the paper is not very well explained nor sufficiently well supported by literature. Is balance just a question of how much time somebody spends in a place per day, or does it have to do with what people exactly are doing there, including spending money or making noise, polluting and so forth. Obviously, with Telco data not much can be said about all this.
Response: We highly appreciate for this insightful comment. In our opinion, this comment contains two issues. First, the concept of ‘balance’ needs to be further clarified. The ‘balanced’ urban and tourism development represents the situation that these historical conservation areas obtaining an equilibrium among residents, commuters and tourists. Specifically, the identification of these three groups are achieved through people’s spatio-temproal behaivor patterns based on cellular data. Further explanations have been added in Section 3.3 Celluar Data Processing (Line 196-222).
Second, we agree that current data-driven approach relied on large amount of cellular data cannot reveal the detailed behavioral effects, i.e., what people exactly are doing there, including spending money or making noise, polluting and so forth. Instead, we firstly classify the people into groups, i.e., residents, commuters and tourists following the selecting principles. Based on the hypothesis that the people within one group will share more similar behavioral pattern on local communities, we then evaluate their impact on balance degree of local communities. The similar analytical pattern has been widely applied in a series of recent studies (Furletti et al., 2012; Paraskevopoulos et al., 2013). Although it cannot provide detailed behavior records for everyone, this data-driven approach is still able to reveal a big picture with the help of large-amount of celluar data. Nevertheless, personal motivation and behavioral effects that could be collected via public survey were not involved in current analysis. We hope the co-present of personal motivation, behavioral effects and spatial-temporal distribution can be achieved in the future to promote a more comprehensive analysis. Detailed discussions about this issue have been added in Section 5.4 Limitations and Next Steps (Line 504-519).
References:
Furletti, B., Gabrielli, L., Renso, C., & Rinzivillo, S. (2012, August). Identifying users profiles from mobile calls habits. In Proceedings of the ACM SIGKDD international workshop on urban computing (pp. 17-24).
Paraskevopoulos, P., Dinh, T. C., Dashdorj, Z., Palpanas, T., & Serafini, L. (2013). Identification and characterization of human behavior patterns from mobile phone data. D4D Challenge session, NetMob.
2) Secondly, the definition of tourists and day trippers, also in literature, is not only dependent on time and place but also on motivation. Again, and this is also related with the first point, the interpretations of Telco data are virtually useless when authors take shortcuts when discussing definitions.
Response: Many thanks for this helpful comment. First of all, we have added a detailed explanation on the identification process of tourists (so call non-local tourists) and day trippers (so call local tourists) in Section 3.3 Celluar Data Processing (Line 196-222). Specifically, the dataset of the whole Shanghai was involved first to identify the local residents and non-local tourists for each ID. Further analyses were made to identify the “recreational places” for local residents. By this way, the day trippers who lived in somewhere else of Shanghai and travelled to a specific historical conservation area for recreational acitivties can be identified. Considering the large-scale collection of personal motivation is difficult due to privacy protection concerns, converting the motivation prediction to principles for classification has became a widely-accepted research paradigm for tourism analysis based on cellular data and other sources of new urban data (Su et al., 2020; Grassini, & Dugheri, 2021).
Moreover, it is a pity that personal motivation and behavioral effects were not involved in current analysis. Classical methods, e.g., public survey and in-depth interview, can address this issus well but they are time-costing and can only be applied within a small scale. Large-scale cellular analysis is capable of covering a whole city but misses the concern of personal motivation and behavioral effects. We hope the co-present of personal motivation, behavioral effects and spatial-temporal distribution can be achieved in the future to promote a more comprehensive analysis. This methodological limitation has been added in Line 512-519.
References: Su, X., Spierings, B., Hooimeijer, P., & Scheider, S. (2020). Where day trippers and tourists go: Comparing the spatio-temporal distribution of Mainland Chinese visitors in Hong Kong using Weibo data. Asia Pacific Journal of Tourism Research, 25(5), 505-523.
Grassini, L., & Dugheri, G. (2021). Mobile phone data and tourism statistics: a broken promise?. National Accounting Review, 3(1), 50-68.
Obviously, the whole discussion the authors develop throughout the paper suffers from these two crucial shortcomings. There is, however, plenty of literature available to address these two important issues and I trust the authors will be able to adjust the paper using this. The paper is interesting and innovative enough to spend some more energy in perfectioning it.
Again, we appreciate all your insightful comments. We would like to express our appreciation for the editor and two anonymous reviewers in the acknowledgement for the time and effort you have taken to provide such insightful guidance. It would be our great pleasure to respond to any further questions and comments that you may have.

Reviewer 3 Report
The paper possesses standard scientific apparatus and is well prepared. It is a bit unusual to put the research questions in the 3. chapter (Materials and methods) instead in the Introduction. Despite that, it works well since it is well elaborated.
The main weakness of this study is that it uses data collected in April 2014. As much as the scientific apparatus applied may be correct, such a big time gap may have influence on the possible application of the study results since we don't know if the latest data on tourism flows have changed. This fact in the research cannot be changed, so it is the authors' moral responsibility to decide if the data are still valid.
Some language editing is required, e.g.:
47 - which is difficult to collect large samples within city scale
48/49 - cellular data with the capacity of recording spatiotemporal distribution of might bring - of what?
73/74 -mThese two approaches obtaining limited samples to draw a convincing conclusion.
128 - while the research sites was set
215 - Dots represents
etc.
Also, correct the following:
142/143 - The total income of tourism and related services in 2018 was 50.9 billion - billion $? which currency?
145/146 - twelve historical conservation areas - in Table 1. you are referring to 20 areas! I understand 12 of them are in the city center and the rest in suburban areas but it is not clear here, so correct.
473 - harmonize font size in: As stated by Brown [55], tourism development activity was always a responsive decision process instead of a preplanning outcome.
Author Response
First, we highly appreciate the editor and the two anonymous reviewers taking the time to offer us comments and insights related to the paper. We greatly appreciate the opportunity to extend our thinking and to improve our manuscript. The responses are given according to the sequence of the reviewer’s comments. All revisions have been highlighted as blue in the manuscript.
Reviewer 3
The paper possesses standard scientific apparatus and is well prepared. It is a bit unusual to put the research questions in the 3. chapter (Materials and methods) instead in the Introduction. Despite that, it works well since it is well elaborated.
1) The main weakness of this study is that it uses data collected in April 2014. As much as the scientific apparatus applied may be correct, such a big time gap may have influence on the possible application of the study results since we don't know if the latest data on tourism flows have changed. This fact in the research cannot be changed, so it is the authors' moral responsibility to decide if the data are still valid.
Response: Thanks a lot for pointing out this limitation. We have updated the data source from 2014 to 2018. Aa series of figures and tables have been revised. Nevertheless, the key findings are still consistent after the changing of data source.
Revision: Figure 3 in Page 6, Figure 6 in Page 11, Table 3 in Page 12, Table 4 in Page 14-15, and Table 5 in Page 15 have been revised according to the analysis based on new data sources.
Minor issues:
Some language editing is required, e.g.:
2) 47 - which is difficult to collect large samples within city scale
Response: Many thanks for this helpful comment. We have revised it in Line 46.
“Nevertheless, the distribution of tracking devices is a time-costing and low efficient process. It is also difficult to collect large samples within city scale.”
3) 48/49 - cellular data with the capacity of recording spatiotemporal distribution of might bring - of what?
Response: Many thanks for this helpful comment. We have revised it in Line 48.
“Therefore, cellular data with the capacity of recording spatiotemporal distribution of human behaviour might bring new research potentials for measuring the degree of balance between urban and tourism development.”
4) 73/74 -These two approaches obtaining limited samples to draw a convincing conclusion.
Response: Many thanks for this helpful comment. We have revised it in Line 73.
“These two approaches obtain too limited samples to draw a convincing conclusion.”
5) 128 - while the research sites was set
Response: Many thanks for this helpful comment. We have revised this language problem in Line 127.
“… while the research sites were set.”
6) 215 - Dots represents
Response: Many thanks for this helpful comment. We have revised it in Line 209.
“Dots represent the raw cell locations”
etc.
Also, correct the following:
7) 142/143 - The total income of tourism and related services in 2018 was 50.9 billion - billion $? which currency?
Response: Many thanks for this reminder. We have added unit of currency in Line 142.
“The total income of tourism and related services in 2018 was 50.9 billion CNY (Chinese Yuan)”
8) 145/146 - twelve historical conservation areas - in Table 1. you are referring to 20 areas! I understand 12 of them are in the city center and the rest in suburban areas but it is not clear here, so correct.
Response: Many thanks for this reminder. We have clarified this number in Line 145-146.
“Resources Bureau has published a conservation agenda for twenty historical conservation areas (with twelve of them locate in the city center) to promote a more sustainable approach between tourism and urban development.”
9) 473 - harmonize font size in: As stated by Brown [55], tourism development activity was always a responsive decision process instead of a preplanning outcome.
Response: Many thanks for this helpful comment. We have changed the font in Line 481.
Again, we appreciate all your insightful comments. We would like to express our appreciation for the editor and two anonymous reviewers in the acknowledgement for the time and effort you have taken to provide such insightful guidance. It would be our great pleasure to respond to any further questions and comments that you may have.

Round 2
Reviewer 1 Report
Thank you for addressing my comments and suggestions. There are a few typos in the paper e.g. line 146 "locate", should be "located" ". Please, kindly review the paper for typos etc.